# Integrating Spatial Omics and Deep Learning: Toward Predictive Models of Cardiomyocyte Differentiation Efficiency

**DOI:** 10.3390/bioengineering12101037

**Published:** 2025-09-27

**Authors:** Tumo Kgabeng, Lulu Wang, Harry M. Ngwangwa, Thanyani Pandelani

**Affiliations:** 1Unisa Biomedical Engineering Research Group, Department of Mechanical, Bioresources, and Biomedical Engineering, School of Engineering and Built Environment, College of Science, Engineering and Technology, University of South Africa (UNISA)—Florida Science Campus, Roodepoort 1709, South Africa; 28346416@mylife.unisa.ac.za (T.K.); luluw@ru.is (L.W.); ngwanhm@unisa.ac.za (H.M.N.); 2Department of Engineering, Reykjavik University, 102 Reykjavik, Iceland

**Keywords:** spatial omics, deep learning, cardiomyocyte differentiation, graph neural networks, recurrent neural networks, cardiac regeneration

## Abstract

Advances in cardiac regenerative medicine increasingly rely on integrating artificial intelligence with spatial multi-omics technologies to decipher intricate cellular dynamics in cardiomyocyte differentiation. This systematic review, synthetising insights from 88 PRISMA selected studies spanning 2015–2025, explores how deep learning architectures, specifically Graph Neural Networks (GNNs) and Recurrent Neural Networks (RNNs), synergise with multi-modal single-cell datasets, spatially resolved transcriptomics, and epigenomics to advance cardiac biology. Innovations in spatial omics technologies have revolutionised our understanding of the organisation of cardiac tissue, revealing novel cellular communities and metabolic landscapes that underlie cardiovascular health and disease. By synthesising cutting-edge methodologies and technical innovations across these 88 studies, this review establishes the foundation for AI-enabled cardiac regeneration, potentially accelerating the clinical adoption of regenerative treatments through improved therapeutic prediction models and mechanistic understanding. We examine deep learning implementations in spatiotemporal genomics, spatial multi-omics applications in cardiac tissues, cardiomyocyte differentiation challenges, and predictive modelling innovations that collectively advance precision cardiology and next-generation regenerative strategies.

## 1. Introduction

Understanding cardiac biology and regenerative medicine has undergone a paradigm shift due to the convergence of deep learning techniques with spatial multi-omics technology [1,2]. The integration of advanced artificial intelligence (AI) techniques, specifically Graph Neural Networks (GNNs) and Recurrent Neural Networks (RNNs), with spatially resolved transcriptomics, epigenomics, and multi-modal datasets addresses fundamental challenges in cardiac regeneration [3,4,5,6]. This integration represents a promising approach to decoding the complex spatiotemporal dynamics that govern cardiomyocyte development and differentiation.

Cardiomyocyte differentiation in the adult human heart remains severely limited, with annual renewal rates of only approximately 0.5%, contrasting sharply with the estimated billion or more cardiomyocytes lost following myocardial infarction [7,8,9]. This limited regenerative capacity underscores the critical need for advanced computational approaches that can optimise induced pluripotent stem cell-derived cardiomyocyte (iPSC-CM) differentiation protocols and predict therapeutic outcomes with high precision.

The scope of this review encompasses deep learning applications in single-cell and spatial biology, recent developments in spatial omics technologies applied to cardiac tissues, and predictive modelling techniques for differentiation efficiency in iPSC-CM. Through systematic review of emerging methodologies and their applications, we aim to establish a comprehensive framework for AI-driven cardiac regenerative medicine that bridges computational innovation with clinical translation.

## 2. Research Methods and Materials

This review was conducted in accordance with the Preferred Reporting Items for Systematic Reviews and Meta-Analyses [10] (PRISMA) guidelines to systematically identify, evaluate, and synthesise literature examining the integration of spatial multi-omics technologies with deep learning frameworks in the context of cardiomyocyte differentiation and cardiac regeneration.

Literature searches, delimited to multiple electronic databases including PubMed, Scopus, Web of Science, and IEEE Xplore, were conducted to ensure coverage of peer-reviewed sources. To ensure the most recent and ongoing work was also captured, additional searches were performed in preprint repositories including bioRxiv, medRxiv, and arXiv. Manual searches in high impact journals such as Nature, Science, Cell, and Circulation Research were also undertaken. The review considered publications from January 2015 to December 2025, a timeframe chosen to capture the latest methodological advances in spatial multi-omics and deep learning.

The search strategy combined keywords and Boolean operators to capture publications at the intersection of cardiac regeneration, spatial omics, and artificial intelligence. The primary query included terms such as “cardiac regeneration”, “spatial transcriptomics”, “spatial epigenomics”, “graph neural networks”, “recurrent neural networks”, “cardiomyocyte differentiation”, and “iPSC-CM”. Additional search terms reflected methodological and thematic priorities, including attention mechanisms, explainable AI, and multi-omics integration.

### 2.1. Study Selection and Screening

The selection process followed a three-phase PRISMA workflow. During the identification stage, database searches yielded 462 records, supplemented with manually curated sources from journals and preprints. Screening was conducted independently by two reviewers (K.M. and T.I.) at the title and abstract level using predefined criteria. Inter-rater reliability was assessed using Cohen’s kappa coefficient (κ = 0.867), which indicated almost perfect agreement, supported by a 96.3% percentage agreement rate. Discrepancies (3.7% disagreement rate) were resolved though consensus-based discussion. Following this process, 60 studies were excluded, leaving 402 records eligible for full-text screening. During the eligibility assessment, 314 full-text articles were excluded for reasons including insufficient methodological detail, reliance solely on bulk omics data, non-cardiac tissue focus, or lack of spatial integration. Reasons for exclusion at this stage encompassed:Duplicate methodologies (n = 10)Insufficient incorporation of deep learning components, such as minimal or no use of GNNs/RNNs/attention networks (n = 10)Preliminary results lacking experimental validation or replication (n = 8)Studies lacking spatial resolution and relying solely on bulk omics data (n = 7)Focus on non-cardiac tissues or unrelated biological contexts (n = 6)Insufficient methodological detail impeding reproducibility (n = 6)Small sample sizes interpretability methodologies for AI models (n = 4)Purely theoretical frameworks without accompanying empirical data (n = 4).

### 2.2. Eligibility Assessment and Synthesis

The inclusion criteria encompassed publications from 2015 to 2025 focusing on cardiac tissue analysis and cardiomyocyte biology, with specific emphasis on studies integrating spatial omics technologies and computational methods relevant to cardiomyocyte differentiation or cardiac regeneration. Eligible studies were required to incorporate advanced deep learning architectures such as graph neural networks, recurrent neural networks, or attention-based models. Only peer-reviewed articles and relevant preprints published in English were considered, and full-text availability was mandatory. Exclusion criteria ruled out non-cardiac applications, studies without spatial resolution, purely theoretical work without validation, conference abstracts, editorials, and studies with insufficient sample sizes or lacking methodological transparency.

Quality assurance was maintained through adherence to PRISMA guidelines, dual independent screening, and systematic application of inclusion and exclusion criteria. Data extraction was structured around ten standardised fields, including bibliographic metadata (title, authors, publication year, DOI), methodological details, and justification for inclusion or exclusion. Ultimately, 88 studies met the inclusion criteria and were included in the qualitative synthesis, a PRISMA 2020 flowchart summarising this process is represented in Figure 1.

Literature synthesis focused on methodological innovation, technical reproducibility, and clinical translation potential, with problem-oriented analysis emphasising integration challenges in multi-omics data and comparative analysis evaluating performance metrics across spatial platforms within the 88-study corpus. Studies were critically appraised to identify emerging computational approaches, assess comparative performance across spatial omics platforms, and highlight contributions toward interpretable and explainable AI. The synthesis framework integrated problem-oriented and comparative analysis, ensuring that findings not only mapped the current state of the field, but also identified limitations and future research opportunities.

As the review relied exclusively on published literature, ethical approval was not required. All included studies reported appropriate ethical oversight in their original publications. To ensure transparency and reproducibility, the database-specific search strings, filters, and syntax adaptations are provided in Appendix A, and the full dataset, search results, and analysis scripts have been made publicly available via a dedicated GitHub repository (https://github.com/Tumo505/ReviewPaperPublications-PRISMA (accessed on 18 September 2025)). Data are provided in CSV, TSV, and Excel formats, with UTF-8 encoding.

## 3. Cardiomyocyte Differentiation and Regenerative Medicine

### 3.1. Cardiomyocyte Biology and Development

Contemporary understanding of cardiomyocyte differentiation has been revolutionised through temporal analysis, revealing four distinct stages, each characterised by specific chromatin alteration patterns [11,12]. These stages demonstrate the intricate connection between chromatin structure and gene expression, with the discovery of unique pre-activation chromatin patterns in cardiac muscle genes providing crucial mechanistic insights [12,13]. The transcriptional synergy—defined as the cooperative enhancement of gene expression where the combined effect of multiple transcription factors exceeds their individual contributions [13,14]—between Gata4 and MEIS1 has emerged as a fundamental regulatory mechanism in cardiomyocyte differentiation. This synergy operates through combinatorial binding at shared enhancers, where MEIS1 and GATA-6 physically interact and mutually reinforce each other’s chromatin occupancy [13,14], resulting in increased accessibility and coordinated activation of cardiac gene expression programmes. Meanwhile, Brg1, a chromatin remodelling factor, works in coordination with Polycomb repressive complexes to orchestrate cardiac differentiation through temporal control of enhancer activation and maintenance of lineage-specific chromatin states [13,14].

Recent studies have identified the zinc finger E-box binding homeobox 1 (ZEB1) as a crucial regulator during early cardiomyocyte differentiation, particularly during the critical transition from mesoderm to cardiac mesoderm [11,15,16]. Signalling pathways including FGF, BMP, and Wnt exhibit biphasic roles in cardiomyocyte differentiation, where temporal activation and inhibition prove essential for accurate cardiac fate determination [14,15].

Despite significant advances in differentiation protocols, iPSC-CMs continue to exhibit fundamental limitations that impede clinical translation [16,17,18]. The primary challenge remains their immature phenotype, as iPSC-CMs differ substantially from mature in vivo cardiomyocytes in electrophysiological parameters, contraction patterns, and gene expression profiles. These cells typically display heterogeneous action potential properties, blending atrial, ventricular, and nodal-like phenotypes within single differentiation cultures.

### 3.2. Current Challenges in iPSC-CM Technology

The translation of iPSC-CM technology faces multiple hurdles that limit therapeutic applications. Cardiomyocyte generation in end-stage heart failure occurs at rates 10–50 times lower than in healthy hearts, according to studies employing nuclear bomb test-driven ^14^C methodology [17]. This severely compromised regenerative capacity emphasises the critical need for enhanced differentiation protocols and improved cell delivery strategies.

Contemporary protocols exhibit variable differentiation efficiency across different iPSC lines, significantly hindering technology application. The immature phenotype of iPSC-CMs represents a persistent challenge, characterised by incomplete sarcomere organisation, immature calcium handling, and altered metabolic profiles compared to adult cardiomyocytes [12,19,20,21]. Emerging metabolic maturation approaches combined with epigenetic regulation has demonstrated potential to address these challenges.

## 4. Spatial Multi-Omics in Cardiac Research

### 4.1. Human Developmental Cardiac Datasets

The landscape of spatial cardiac research has been transformed by comprehensive datasets providing unprecedented resolution of the human heart [22,23,24]. The Human Cell Atlas Heart Development project combines single-cell RNA sequencing with spatial transcriptomics to map cardiac development from 5.5 to 14 weeks post-conception, revealing temporal dynamics of cellular specification and spatial organisation [23,24,25].

The Spatial Dynamics of Developing Human Heart dataset employs MERFISH technology to achieve subcellular resolution mapping between 9 and 16 weeks post-conception [24,26]. This resource identifies 75 distinct cell states, including previously unrecognised resident chromaffin cells and cardiac cell types [24,26]; the dataset provides crucial insights into multicellular signalling mechanisms coordinating spatial organisation during cardiac morphogenesis and demonstrates the complex laminar organisation of ventricular cardiomyocyte subpopulations across ventricular wall [23,24,27].

The Adult Human Cell Atlas represents the most comprehensive characterisation of adult human heart cellular diversity, analysing approximately 500,000 cells from six anatomical regions [24,28]. This resource reveals distinct atrial and ventricular fractions with diverse developmental origins while highlighting cellular heterogeneity among cardiomyocytes, pericytes, and fibroblasts [24,28]. The atlas has become an essential resource for understanding normal cardiac function and disease mechanisms [28].

### 4.2. Regenerative Model Systems

The Zebrafish Heart Regeneration Atlas provides the most extensive spatial-temporal resource for cardiac regeneration studies, encompassing 569,859 spots across eight regeneration stages [23]. This dataset integrates single-cell RNA sequencing with spatial transcriptomics (Stereo-seq) to characterise cardiomyocyte cell state transitions leading to regenerated myocardium development [23]. The resource creates a 4D “virtual regenerating heart” that acts as an invaluable tool for cardiovascular regeneration research [23].

The development of iPSC-derived cardiac organoids now provides novel platforms for investigating human cardiac development in controlled settings [24,28]. Human heart organoid spatial atlases demonstrate temporal progression from initial seeding through 20 days of differentiation, providing insights into multicellular organisation and intercellular communication networks that govern cardiac morphogenesis [24,28].

Dataset Harmonisation Challenges:
bioengineering-12-01037-t001_Table 1Table 1The datasets reviewed herein encompass a broad spectrum of species, tissue types, molecular modalities, and temporal scales, collectively constituting a comprehensive resource for understanding cardiac development, physiological homeostasis, pathological remodelling, and regeneration. They include data derived from human embryonic and adult hearts, infarcted myocardial tissue, cardiac conduction system samples, iPSC-CMs, cardiac organoids, and regenerating zebrafish hearts. At the molecular level, these datasets leverage cutting-edge single-cell and spatial technologies, such as single-cell RNA sequencing (scRNA-seq), single-nucleus RNA sequencing, and single-cell and single-nucleus ATAC sequencing (scATAC-seq and snATAC-seq), along with diverse spatial transcriptomics platforms including 10x Visium, MERFISH, Stereo-seq and in situ sequencing (ISS). Multimodal integrative approaches are prominent, enabling concurrent interrogation of transcriptomic and epigenomic landscapes within spatially resolved cellular contexts.SubcategoryDatasetModalityTissueTemporal InfoIntegration ConsiderationsCitationDevelopmentalHuman Cell Atlas Heart DevelopmentscRNA-seq + Spatial TranscriptomicsHuman embryonic heartYes (5.5–14 Weeks Post Conception (WPC))Different platforms (10x Visium vs. scRNA-seq) require cross-modal anchoring (e.g., Seurat v4 WNN, Harmony) to align spatial and transcriptomic resolution. Challenges include batch effects from donors or technologies, potential loss of spatial details during integration, computational scalability for large datasets (e.g., >500,000 cells), and ensuring accurate cell type deconvolution without over-smoothing heterogeneous populations. These issues impact the reliability of organ-wide atlases by introducing artefacts in cellular interaction mapping.[29]
Spatial dynamics of developing human heartscRNA-seq + MERFISHHuman embryonic heartYes (9–16 WPC)MERFISH has higher spatial resolution but limited gene coverage vs. scRNA-seq; integration often requires feature selection (e.g., shared genes) + imputation methods using methods like Tangram (probabilistic mapping to minimise divergence), gimVI (deep generative joint modelling), or Spatialscope (score-based diffusion models with Potts spatial smoothness).Limitations include resolution mismatch (MERFISH vs. Visium-like spot-based aggregation, leading to potential loss of fine details), over-smoothing of spatial heterogeneity, high computational costs, dependency on scRNA-seq reference quality (poor quality introduces artefacts), and batch effects requiring correction.[30]
Developing human heart (EGA)snRNA-seq + Spatial Transcriptomics + ISSHuman embryonic heartYes (4.5, 6.5, 9 WPC)ISS vs snRNA-seq differ in throughput and detection sensitivity; anchor-based correction recommended (e.g., Seurat anchors for label transfer). Challenges include varying detection rates leading to incomplete gene profiles, batch effects from multi-modal data sources, normalisation difficulties for low-abundance transcripts, and integration of imaging-based ISS with sequencing data without losing spatial precision. These issues can hinder accurate reconstruction of early cardiac development trajectories, potentially introducing biases in cell state identification.[31]
Mouse Heart Spatiotemporal Atlas (Stereo-seq)Spatial Transcriptomics (Stereo-seq)Mouse heartYes (embryonic day 20 (E20), postnatal day 1 (P01), postnatal day 4 (P04), postnatal day 14 (P14))Stereo-seq has ultra-high resolution; batch alignment needed for cross-species inference (human vs. mouse) using ortholog mapping and methods like Harmony or MNN. Challenges involve handling large data volumes (>500,000 spots), accuracy of ortholog mapping across species, potential over-smoothing in dimensionality reduction, and temporal batch effects from multiple developmental stages. These limitations impact comparative analyses with human data, risking misinterpretation of conserved cardiac organogenesis mechanisms.[32]
Mouse heart spatial transcriptomics (Visium)scRNA-seqiPSC-derived cardiomyocytesYes (pluripotency (day 0), germ layer specification (day 2), progenitor cardiac cell state (day 5), committed cardiac cell state (day 15), definitive cell state (day 30))Integration across iPSC protocols requires batch-effect correction (MNN, LIGER) due to lab-specific variability. Challenges include variability in differentiation efficiency leading to heterogeneous cell states, data sparsity in scRNA-seq, ensuring accurate cell type mapping without reference overfitting, and handling temporal trajectories with potential dropout events. These issues affect modelling of cardiomyocyte maturation, potentially leading to biassed predictions of protocol outcomes.[33]
Human heart organoids spatial atlasscRNA-seq + Spatial transcriptomicsHuman heart organoidsYes (approximately 3 days after seeding (day 0)–day 20 of differentiation)Organoids differ from in vivo tissues in cellular composition; transfer learning-based integration may be required (e.g., using pre-trained models from in vivo data). Challenges encompass discrepancies in cell maturity and states between organoids and native tissues, limited spatial resolution in miniaturised models, batch effects from culture conditions, and validation against human samples to avoid artefactual networks. These limitations influence insights into morphogenesis, risking overgeneralisation from in vitro to in vivo contexts.[34]
Human SAN Cell AtlasscRNA-seq + scATAC-seqHuman sinoatrial node (iPSC)Yes (Differentiation)Requires multi-modal alignment (RNA + ATAC); weighted nearest neighbour (WNN) or MOFA+ commonly applied for joint embedding. Challenges include differing data sparsity (ATAC more sparse than RNA), accuracy of peak-to-gene linking, computational demands for integrating epigenetic and transcriptomic layers, and handling differentiation-induced variability. These issues impact pacemaker cell identification, potentially introducing errors in regulatory network inference.[35]
Adult (Physiological)
Adult human heart cell atlasscRNA-seq + snRNA-seqAdult human heartNo (Adult)snRNA-seq vs scRNA-seq differ in transcript detection bias (nuclear vs. cytoplasmic); normalisation across modalities essential (e.g., using SCTransform). Challenges involve lower gene detection in snRNA-seq, integration without losing rare cell types, batch effects from anatomical regions, and ensuring comparability in large-scale atlases (>500,000 cells). These limitations affect cellular heterogeneity mapping, risking underrepresentation of dynamic states in healthy hearts.[36]
Spatially resolved multiomics of human cardiac nichesscRNA-seq + snATAC-seq + Spatial TranscriptomicsAdult human heartNo (Adult)Multi-omics alignment requires matrix factorisation or LIGER for shared latent space inference. Challenges include integrating three modalities with varying resolutions (spatial vs. single-cell), batch variations from donors or regions, preserving spatial context in multi-omic inference, and handling sparsity in ATAC data. These issues influence niche discovery, potentially leading to incomplete cellular interaction models[37]
Human cardiac conduction systemscRNA-seq + Spatial TranscriptomicsHuman cardiac conduction systemNo (Adult)Cell type resolution differs across datasets; label transfer + cross-modal anchoring recommended. Challenges encompass aligning conduction-specific markers, handling low-abundance pacemaker cells, spatial deconvolution accuracy in heterogeneous tissues, and batch effects from sample preparation. These limitations affect understanding of electrical signalling, risking misattribution of cell roles.[6,37,38]
Pathological/Regenerative
Spatial multi-omic map of human MI (Myocardial Infarction)snRNA-seq + snATAC-seq + Spatial TranscriptomicsHuman infarcted heartYes (Post-MI timepoints)Post-MI inflammatory environments induce batch-specific effects; regression-based correction (e.g., Harmony) improves comparability. Challenges include disease-induced heterogeneity complicating alignment, integrating chromatin/epigenetic with spatial data, temporal variability across MI stages, and sparsity in infarct zones. These issues impact remodelling maps, potentially biassing therapeutic target identification.[33]
Human heart spatial transcriptomics (Disease)Spatial TranscriptomicsHuman heart (disease)No (Disease states)Differences in sample preparation (frozen vs. FFPE) require careful normalisation (e.g., using sctransform or DESeq2). Challenges involve tissue quality variations in diseased samples, artefact removal from pathology-induced noise, ensuring comparability across disease states, and handling low-resolution spots in heterogeneous lesions. These limitations affect disease progression modelling, risking inaccurate spatial gene expression profiles.[31,39]
Zebrafish heart regeneration atlasscRNA-seq + Spatial Transcriptomics (Stereo-seq)Zebrafish heartYes (8 timepoints of zebrafish heart regeneration stages)Cross-species integration requires ortholog mapping + dimensionality reduction alignment. Challenges include species-specific gene expression differences, handling high-resolution Stereo-seq data volumes, temporal alignment across regeneration stages, and batch effects from injury timepoints. These issues influence comparative regenerative studies, potentially leading to translational gaps with human models[40]
iPSC-CM scRNA-seq + scATAC-seqscRNA-seq + scATAC-seqiPSC-derived cardiomyocytesYes (Day 0–30)Multi-modal integration (RNA + ATAC) typically handled via Seurat v4 WNN or scGLUE for joint analysis. Challenges encompass sparsity in ATAC-seq data, linking enhancers to genes accurately, variability in iPSC differentiation trajectories, and computational scaling for time-series data. These limitations affect maturation dynamics insights, risking biassed regulatory network reconstructions.[41]

While Table 1 presents a comprehensive collection of cardiac datasets, critical challenges exist in harmonising data across different platforms and experimental conditions that significantly impact cross-study inference [42,43,44].

Platform-Specific Variability:

Different spatial transcriptomics platforms exhibit substantial technical differences:Resolution Heterogeneity: Visium (55 μm spots), MERFISH (subcellular), and Stereo-seq (500 nm–1 μm) platforms operate at vastly different spatial scales [26,42].Gene Coverage Variability: Platform-specific gene panels range from ~18,000 genes (Visium) to targeted panels of 100–1000 genes (MERFISH, SeqFISH+) [26,43].Sensitivity Differences: Detection efficiency varies significantly across technologies, with some platforms showing higher dropout rates for lowly expressed genes [26,44].

Batch Effects and Correction Strategies:

Systematic batch effects arise from multiple sources:Technical Variability: Differences in tissue preparation protocols, sectioning thickness (4–20 μm), and storage conditions introduce substantial noise [43,44].Temporal Effects: Sample collection timing, processing delays, and storage duration create systematic biases [43].Laboratory Effects: Inter-laboratory variations in reagent preparation, instrumentation calibration, and operator expertise [44].

Integration Methodologies:

Current harmonisation approaches show limited effectiveness:Mutual Nearest Neighbour (MNN) Correction: While successful for single-cell RNA-seq integration, MNN approaches struggle with spatial data due to coordinate system differences and varying cell type distributions [45,46].Cross-Modal Anchoring: Identification of shared cellular landmarks across platforms remains challenging due to different feature representations and measurement scales [42,45].Batch Effect Correction: Methods like Harmony and RPCA show promise but require careful parameter tuning and may overcorrect biological signals [44,47].

These harmonisation challenges necessitate platform-specific validation and limit the direct comparability of results across studies, potentially compromising meta-analyses and systematic reviews [42,43,48].

## 5. Deep Learning Architectures for Spatial Cardiac Data

### 5.1. GNNs in Cardiac Applications

Graph Neural Networks have emerged as powerful tools for analysing spatial relationships in cardiac tissues, where cellular interactions and spatial organisation critically determine function [48,49]. GNNs excel at capturing complex cell–cell communication patterns and spatial dependencies that traditional analytical approaches struggle to model effectively. The comparative performance metrics of the selected deep learning architectures is detailed in Table 2.

Adaptive graph models such as spaCI (schematic overview shown in Figure 2) incorporate spatial coordinates with gene expression data to effectively detect active ligand–receptor signalling between neighbouring cells [50]. These approaches demonstrate superior performance compared to existing methods on both simulated and real-world spatial transcriptomics datasets, providing novel insights into upstream transcriptional factors mediating active cellular interactions [50]. The integration of triplet loss functions and an attention mechanism enables robust inference of cellular communications from sparse spatial data [50]. 

Because GNNs naturally depict cellular linkages and geographical interactions, they have completely changed the way spatial data is being analysed [51]. The primary benefit of GNNs is their capacity to concurrently represent spatial neighbourhood information and gene expression similarity, making them perfect for spatially resolved transcriptomics analysis [52,53]. The effectiveness of GNN-based approaches for cell-type deconvolution and spatial domain identification has been shown by the recent applications in spatial transcriptomics [51]. Real spots can be adapted from pseudo-spots by dual graph construction employing expressive and spatial information according to the STdGCN framework, a novel graph neural network architecture, which incorporates expression profiles from single-cell data and spatial localisation information from spatial transcriptomics data for cell type deconvolution [54]. Extensive benchmarking demonstrates STdGCN’s superiority over 14 published state-of-the-art models, with successful applications in human breast cancer and heart development studies [54]. This approach has demonstrated exceptional effectiveness in predicting cell-type proportions across a variety of spatial transcriptomics platforms [48,54]. Spatial transcriptomic cell-type deconvolution represents another critical application where GNNs demonstrate exceptional performance. The schematic view of STdGCN framework is shown in Figure 3.
bioengineering-12-01037-t002_Table 2Table 2Comparative performance metrics of selected deep learning architectures applied to cardiac and spatial transcriptomics datasets. This table presents key evaluation metrics, including accuracy, area under the receiver operating characteristic curve (AUROC), F1 score, precision, and recall for multiple models such as STdGCN, spaCI, LSTM, Graph Transformer (GT), Random Forest baseline, and hybrid GNN-LSTM architectures across diverse cardiac-related datasets. Some metrics are unavailable (-) due to differences in reporting across studies. The ‘Baseline Comparison’ column specifies the baseline models used for each model’s evaluation. For spaCI, evaluation was conducted against a Random Forest baseline and a standard Graph Convolutional Network (GCN) on four spatial transcriptomics cohorts, each comprising single-cell RNA sequencing data from human breast cancer tissues (Cohort 1: n = 100, Cohort 2: n = 150, Cohort 3: n = 120, Cohort 4: n = 130). spaCI’s performance reflects its ability to identify true ligand–receptor interaction pairs, with metrics reported as mean ± standard error (SE) across cohorts. The Random Forest baseline serves as a traditional benchmark for heart failure prediction tasks, using electronic health record (EHR) data from heart failure patients (n = 200). The hybrid GNN-LSTM model achieves high accuracy in heart failure prediction, exemplifying the potential of integrated spatiotemporal approaches. Performance values were adapted from the source studies in the final column [50,54,55,56,57,58,59,60,61].Model ArchitectureDatasetAccuracy (%)AUROCF1 ScorePrecisionRecallBaseline ComparisonReferencesSTdGCNHuman breast cancer & heart development-0.920.85--RCTD, SPOTlight, Cell2location, DSTG, CARD[54]
spaCI (GNN+attention)Spatial transcriptomics-0.82Accurately identified true interaction pairs across 4 cohortsCohort 1: mean ± SE: 0.852 ± 0.014Cohort 2: 0.817 ± 0.05Cohort 3: 0.859 ± 0.06Cohort 4: 0.853 ± 0.03--iTALK, CellPhoneDB, CellChat, Connectome[50]
LSTM (cardiac prediction)EHR cardiac data-0.76---Logistic Regression, Naïve Bayes[55,56,57]
Graph Transformer (GT)
Heart failure prediction-0.79250.5361--Random Forest, GraphSAGE, GAT[58]
Random Forest (baseline)
Heart failure prediction0.910.900.910.920.90None (serves as baseline, compared to XGBoost, KNN, logistic regression)[59]
GNN-LSTM (hybrid)
Heart failure prediction98.90----Random Forest, LSTM[60,61,62]

### 5.2. RNNs for Temporal Modelling

Recurrent Neural Networks have shown potential in capturing temporal dynamics in cardiac differentiation and development. These architectures excel at capturing sequential dependencies and temporal patterns that characterise cardiac cell fate decisions and differentiation trajectories. The integration of attention mechanisms with RNN architectures enables the identification of critical temporal windows that influence cardiac differentiation outcomes. Recent studies have shown the application of dual attention RNN models to predict gene temporal dynamics from synthetic time series data derived from gene regulatory networks [63,64]. These models produce remarkably accurate predictions across various network architectures, while the attention mechanism offers comprehensible insights on gene interactions [65,66,67]. Different gene regulatory network architectures can be distinguished hierarchically using graph theory to analyse attention patterns [64].

Predicting cell–cell interactions from spatial transcriptomics data has been explored using the combined capabilities of Long Short-Term Memory (LSTM) and GNNs [60,61]. This hybrid approach leverages the graph-based potential of GNNs for spatial context modelling and the sequence learning capabilities of LSTM [68,69]. When compared to conventional methods, rigorous testing has shown improved predictive capabilities, with backwards search integration producing optimal performance [70].

Moreover, LSTM networks have been used to predict protein subcellular localisation from DNA sequences, outperforming earlier benchmark models that lacked human-engineered features in terms of accuracy [44,71,72]. The versatility of RNN architectures in genomics applications has been demonstrated by the successful resolution of sequence-based biological problems through the integration of convolutional layers between raw data and LSTM inputs [42,43,45].

### 5.3. Integrated Spatiotemporal Architectures

The combination of GNNs and RNNs creates powerful hybrid architectures capable of modelling both spatial organisation and temporal dynamics simultaneously [49,73,74]. These integrated approaches represent the emerging area of computational cardiac biology, enabling comprehensive analysis of how spatial cellular organisation evolves during development and disease progression [47,48]. However, these hybrid architectures require complex interpretability frameworks to elucidate their decision-making processes [48,49]. Traditional explainable AI techniques need significant modification to handle spatiotemporal learning systems operating on irregular graph structures with temporal dependencies [75,76]. The development of such interpretability frameworks remains an active area of research with crucial implications for clinical translation.

### 5.4. Explainability Challenges

Current hybrid architectures suffer from substantial interpretability limitations. Unlike traditional attention mechanisms that provide some insight into model focus, the complex interactions between graph convolutions and recurrent layers create “black box” effects that are difficult to interpret clinically [75,76]. While attention heatmaps and saliency maps have been developed for individual GNN and RNN components, their combination in hybrid architectures requires novel explainable AI frameworks that remain largely undeveloped [75,76].

### 5.5. Real-World Validation Limitations

Most reported performance metrics derive from controlled experimental datasets that may not reflect clinical complexity [77,78]. Key challenges include:
Graph Irregularity: Spatial cellular graphs exhibit irregular topologies that complicate model generalisation across different tissue preparations and imaging platforms [24,79].Temporal Noise: Real-world temporal data contains significant measurement artefacts and missing values that can substantially degrade model performance [80,81,82,83].Cross-Dataset Generalisation: Models trained on specific experimental conditions often fail to generalise to different iPSC lines, differentiation protocols, or laboratory conditions [84,85].Scale Limitations: Current validation studies typically involve small sample sizes insufficient for robust statistical inference about clinical utility [86,87].

These limitations suggest that while hybrid GNN-RNN approaches represent a promising research direction, substantial methodological advances in interpretability and validation are required before clinical translation [88,89,90,91,92,93].

## 6. Predictive Modelling for Differentiation Efficiency

### 6.1. AI Approaches for Predicting Differentiation Outcomes

Comparative analyses of supervised versus unsupervised learning approaches for differentiation prediction have demonstrated the critical importance of labelled datasets in achieving optimal results [46,94,95]. Cross-validation methodologies and holdout validation approaches ensure robust evaluation of model performance across diverse experimental settings and datasets [96]. This incorporation of feature selection methods like Lasso regularisation has enhanced model efficiency while preventing overfitting in high-dimensional biological datasets [95,97]. Multimodal data integration has become a focal point in machine learning models aimed at improving iPSC-CM differentiation prediction models [97]. Single-cell RNA sequencing analysis has revealed key regulators and differentiation trajectories of iPSC-derived cardiomyocytes, identifying candidate genes including CREG and NR2F2 that play important regulatory roles in cardiomyocyte lineage commitment [95].

Advanced computational approaches now enable the prediction of differentiation efficiency based on early-stage molecular signatures [95]. These predictive models incorporate epigenetic modifications, transcriptional dynamics, and metabolic indicators to predict differentiation outcomes with increasing accuracy. The integration of real-time monitoring technologies with predictive algorithms enables dynamic protocol optimisation and improved reproducibility across different iPSC lines [97].

### 6.2. Model Evaluation and Validation Strategies

Comprehensive evaluation frameworks for AI-based differentiation prediction must address the unique challenges posed by biological variability and experimental heterogeneity [96]. Reliability diagrams provide a visual assessment of model calibration by plotting observed event frequencies against predicted probabilities [96]. Well-calibrated models demonstrate agreement between predicted probabilities and observed frequencies, with ideal reliability diagrams displaying function relationships [96].

Reliability diagram methodologies have advanced through the use of the CORP (Consistent, optimally binned, Reproducible, Pooled-adjacent violators) technique, which optimally bins data and employs isotonic regression for robust statistical assessment [96,98]. These diagrams effectively identify systematic biases in probability estimations, revealing whether models consistently overestimate or underestimate differentiation success likelihood, e.g., a model might predict a 70% probability of successful cardiomyocyte differentiation for a cell type that empirically succeeds only 50% of the time, providing crucial information for clinical translation and experimental design optimisation [98,99].

## 7. Limitations and Future Directions

Current limitations in spatial omics technologies include resolution constraints, throughput limitations, and standardisation challenges across platforms [100,101]. Emerging high-resolution spatial transcriptomics technologies aim to overcome current limitations, achieving subcellular resolution without sacrificing genome-wide coverage [102,103]. The integration of multiple omics modalities presents significant computational challenges, requiring sophisticated algorithms capable of handling data types and scales [100,103]. Future development must focus on creating standardised pipelines and benchmarking frameworks that enable reproducible and comparable analyses across different studies and platforms [104,105].

The translation of AI-driven cardiac regeneration technologies to clinical applications requires addressing several critical challenges, including regulatory approval pathways, safety validation, and scalability concerns [106,107]. Improvements in good manufacturing practice (GMP) protocols for iPSC-CM production and advances in cell delivery technologies suggest that clinical translation may be achievable within the next decade [108,109]. Personalised medicine approaches leveraging patient-specific iPSCs combined with AI-guided differentiation protocols represent a promising avenue for addressing individual variability in treatment responses [110,111]. The development of companion diagnostic tools and predictive biomarkers will be essential for successful clinical implementation [112].

## 8. Conclusions

The integration of spatial omics technologies with deep learning approaches represents a transformative paradigm in cardiac regenerative medicine. Graph Neural Networks and Recurrent Neural Networks provide powerful frameworks for analysing the complex spatial and temporal dynamics governing cardiomyocyte differentiation and cardiac development. Recent advances in spatial transcriptomics, epigenomics, and multi-modal datasets have created unprecedented opportunities for understanding cardiac biology at molecular resolution. The convergence of these technologies has enabled significant progress in predictive modelling for differentiation efficiency, spatial analysis of cardiac tissues, and mechanistic understanding of regenerative processes. However, substantial challenges remain in developing robust interpretability frameworks, standardising analytical pipelines, and translating computational insights to clinical applications. Future research should prioritise the development of integrated spatiotemporal architectures that can simultaneously model spatial organisation and temporal dynamics, the creation of standardised benchmarking frameworks for comparative evaluation, and the establishment of regulatory pathways for AI-guided therapeutic development. The successful integration of these approaches promises to accelerate the clinical translation of cardiac regenerative therapies and advance precision medicine approaches for cardiovascular disease treatment. The field stands at a critical juncture where technological capabilities are beginning to match the complexity of biological systems. Continued innovation in computational methods, combined with advances in spatial omics technologies and stem cell biology, will drive the next generation of breakthroughs in cardiac regenerative medicine. Success will require collaborative efforts across disciplines, standardisation of methodologies, and sustained investment in both basic research and clinical translation initiatives.

## Figures and Tables

**Figure 1 bioengineering-12-01037-f001:**
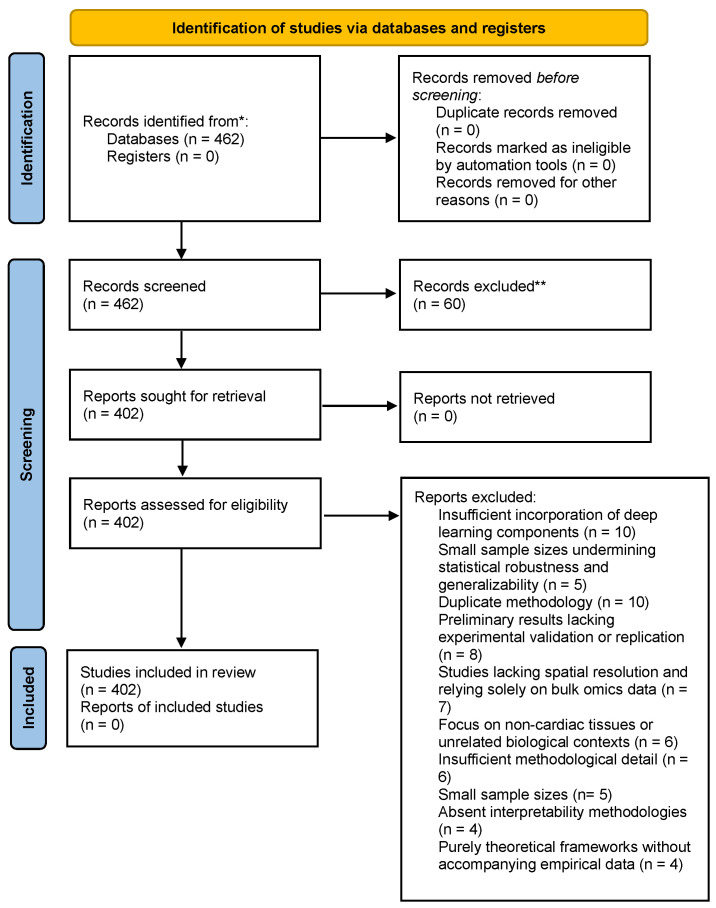
PRISMA flowchart summarising the process followed in selected the studies included in this review. “*” denotes records identified, and “**” denotes records excluded.

**Figure 2 bioengineering-12-01037-f002:**
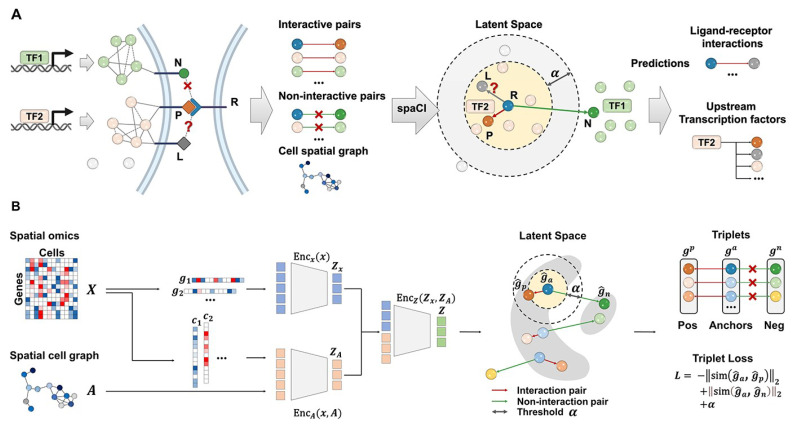
Schematic overview of spaCI. (**A**) The spaCI methodology exploits both the spatial organisation of cells and comprehensive gene expression profiles to infer ligand–receptor (L-R) interactions within tissue contexts. (**B**) In this framework, gene features are embedded into a latent representation via two complementary components: a gene-centric linear encoder and a cell-centric attentive graph encoder. This approach enables simultaneous integration of transcriptional patterns and spatial cellular proximities within the latent space [50].

**Figure 3 bioengineering-12-01037-f003:**
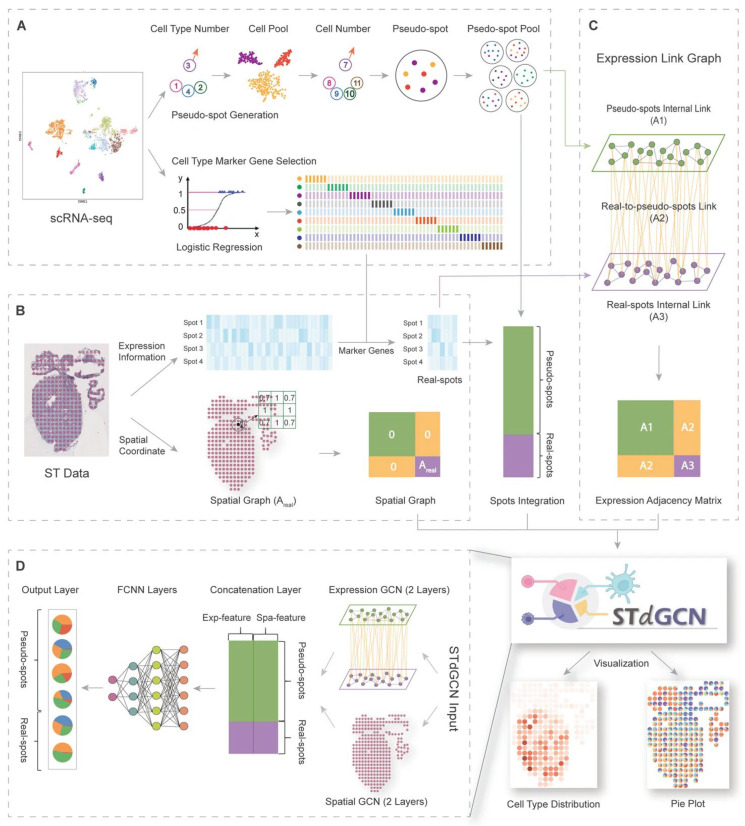
Schematic view of the STdGCN framework. Initially, STdGCN utilises scRNA-seq reference profiles to identify marker genes specific to cell types and to construct a pseudo-spot pool (**A**); it then constructs two distinct graphs: a spatial graph representing spatial relationships among spots (**B**) and an expression graph (**C**); finally, these graphs are integrated as input into the STdGCN model (**D**), where fully connected neural network (FCNN) layers process expression features, spatial graph convolutional network (GCN) layers capture spatial dependencies, and expression GCN layers extract gene expression relationships. The outputs are concatenated and passed through the prediction layers, enabling inference of cell type composition within spots and generation of interpretable visualisations. [54].

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
