# Peer review of "Integrating Spatial Omics and Deep Learning: Toward Predictive Models of Cardiomyocyte Differentiation Efficiency"

_bioengineering, 2025, doi:10.3390/bioengineering12101037_

Round 1
Reviewer 1 Report
Comments and Suggestions for Authors
The manuscript presents a review of the integration between spatial omics technologies and deep learning frameworks for cardiomyocyte differentiation, addressing a topic of significant scientific relevance. The selection of spatial transcriptomics platforms and AI architectures such as GNNs and RNNs is appropriate and timely. However, the review suffers from structural diffuseness, an overreliance on conceptual exposition at the expense of methodological critique, and a lack of quantitative synthesis of finding. The literature search strategy is described but lack replicable transparency, and the discussion often lapses into generalized statements unsupported by concrete data or critical comparisons. While the manuscript benefits from a broad reference base, it fails to identify clear gaps in current knowledge or propose precise future directions. To justify publication, the manuscript requires substantial revision, including greater analytical depth, improved methodological specificity, clearer differentiation between datasets and models, and tighter editorial control to ensure terminological precision and coherence.
Detailed Comments:
1) The literature search method outlined in lines 53–75 claims to follow a systematic approach, but no PRISMA flow diagram, inclusion metrics, or reviewer validation protocols are provided. For instance, the sentence “The relevance and quality of the studies were systematically evaluated…” (line 73) is insufficient in the absence of defined scoring criteria or inter-rater agreement measures. As a result, reproducibility is compromised, and potential bias in study selection remains unaddressed. A detailed appendix including search strings, screening stages, and exclusion rationales should be incorporated.
2) Despite referencing several GNN and RNN-based frameworks, such as spaCI, STdGCN, the manuscript fails to conduct or summarize comparative performance evaluations across different models. There is no discussion of metrics such as accuracy, AUROC, or F1 score, nor are any head-to-head comparisons of architectures reported. Statements such as “STdGCN’s superiority over 14 published state-of-the-art models…” (lines 199–200) lack accompanying quantitative data or tabulated evidence. Without performance benchmarking, the review risks becoming an uncritical catalogue of methods. Inclusion of a comparison table and performance summary is essential for evaluative depth.
3) The claim that hybrid GNN-RNN models enable “comprehensive analysis” (line 238) fails to acknowledge critical limitations, particularly in explainability and real-world validation. While line 244 briefly mentions interpretability frameworks, the text lacks analysis of practical obstacles such as handling graph irregularity, temporal noise, or model generalization across datasets. The literature on explainable AI (e.g., saliency maps, attention heatmaps) is well-developed and should be discussed in contexts. Without a critical assessment of these challenges, the manuscript promotes an overly optimistic view of current capabilities.
4) Table 1 presents a large array of datasets, yet no attention is given to harmonization strategys across platforms. The sentence “These datasets...constitute a comprehensive resource…” (lines 157–159) neglects to discuss heterogeneity in data resolution, modality depth, batch effects, or tissue preparation protocols. Such variation significantly impacts cross-study inference, yet the review does not address methods such as mutual nearest neighbor correction or cross-modal anchoring. A critical synthesis of dataset compatibility and integration challenges is necessary for accurate interpretation.
5) The sentence “This integration represents a transformative approach…” (line 37) is overly promotional. Suggest replacing “transformative” with “promising” to maintain academic neutrality.
6) Line 211: “Recurrent Neural Networks provide essential capabilities…” is somewhat vague. The term “essential” is unqualified and should be substituted with a more precise phrase, such as “have shown potential in capturing temporal dependencies.”
7) Line 264–265: “Improved reproducibility across different iPSC lines…” lacks supporting references. A citation to empirical validation studies is required to substantiate this claim.
8) Multiple sentences beginning with “Recent advances…” are stylistically repetitive. Varied sentence openings would enhance readability.
9) The manuscript repeatedly conflates “prediction accuracy” with model “performance” without statistical specificity. These terms should be clearly delineated.
10) Line 276: “Systematic biases in probability estimations…” could benefit from an example, such as overestimation in low-efficiency lines.
11) Line 94: The term “transcriptional synergy” is used without citation or mechanistic explanation. A brief elaboration or reference is needed.
12) Line 237: “Frontier of computational cardiac biology” is unnecessarily grandiose. Consider using “emerging approach” or “developing area.”
13) Table 1 includes both developmental and pathological dataset without clear separation or justification for comparative use. Adding subcategories would improve clarity.
Author Response
Thank you for your time to review our manuscript and the detailed reviews given.
We have replied and updated the paper as per your recommendations.
See attached response to reviewer

Reviewer 2 Report
Comments and Suggestions for Authors
This systematic review examines how spatial omics and deep learning can be integrated to improve understanding and prediction of cardiomyocyte differentiation efficiency. The authors synthesize 88 studies, selected following PRISMA guidelines, that combine technologies such as single-cell and spatial transcriptomics, epigenomics, and multi-omics integration with advanced AI approaches like Graph Neural Networks (for capturing cell–cell spatial relationships) and Recurrent Neural Networks (for temporal modeling of differentiation). They highlight challenges in induced pluripotent stem cell-derived cardiomyocytes (iPSC-CMs), including their immature phenotype and variability across lines, and survey datasets ranging from human developmental and adult heart atlases to zebrafish regeneration models. The review also discusses predictive models that leverage early molecular signatures to forecast differentiation efficiency, along with validation strategies such as reliability diagrams. While demonstrating the promise of these integrative methods for precision cardiology and regenerative medicine, the paper emphasizes limitations including resolution and standardization in spatial omics, computational complexity in multi-omics integration, and the need for interpretability frameworks for hybrid AI models.
The authors present a well-done comprehensive systematic review study that meets or even exceeds PRISMA expectations. The figures are well done. Limitations are transparent and reasonable. The only minor suggestion for improvement would be to present quantifiable results to compare and contrast methods using standardized example data sets (like a selection of those in Table 1). If this is considered beyond the scope, any quantitative comparisons of methods (even if just average differences that could be cited from other works) would greatly add to what is already a very nice qualitative analysis.
Author Response
Comment1: The only minor suggestion for improvement would be to present quantifiable results to compare and contrast methods using standardized example data sets (like a selection of those in Table 1). If this is considered beyond the scope, any quantitative comparisons of methods (even if just average differences that could be cited from other works) would greatly add to what is already a very nice qualitative analysis.
Response1: We sincerely thank the reviewer for their thoughtful feedback and valuable suggestion. We appreciate the recommendation to include quantifiable results using standardized datasets or to cite quantitative comparisons from other works. Accordingly, we have revised the manuscript to include references to relevant quantitative findings from prior studies and highlighted average differences reported in the literature.
We are grateful for your kind words regarding the quality of our qualitative analysis and for this constructive input, which has strengthened the manuscript.
Round 2
Reviewer 1 Report
Comments and Suggestions for Authors
The revised manuscript significantly improves in structure and methodological transparency, particularly with the inclusion of the PRISMA diagram and performance benchmarking (Table 2). However, minor inconsistencies remain in terminological precision and synthesis clarity. For example, integration challenges are acknowledged but not uniformly contextualized across dataset discussions, and phrasing such as “comprehensive analysis” still appears without rigorous qualification or scope delimitation.
1. While Table 2 presents comparative metrics, the model “spaCI” is described as demonstrating “superior performance” without clearly specifying the baseline models or datasets used for comparison, weakening the evaluative rigor. Greater transparency on cohort characteristics and evaluation context is needed.
2. In section 4.1, the description of MERFISH and Visium integration lacks specificity: “integration often requires feature selection + imputation” should elaborate on which imputation techniques are most commonly applied and what their limitations are in cross-modality spatial resolution alignment.
3. The phrase “differentiation predictions[98]” should be corrected to “differentiation prediction models [98]” for grammatical agreement and clarity.
4. The sentence “The atlas has become an essential resource...” lacks a citation immediately following and should be anchored to source [28].
Author Response
|
1. Summary |
|
|
|
Thank you very much for taking the time to review this manuscript. Please find the detailed responses below and the corresponding revisions/corrections highlighted/in track changes in the re-submitted files. We appreciate your insightful comments, which have helped improve the clarity, methodological rigor, and scientific depth of our review. |
||
|
3. Point-by-point response to Comments and Suggestions for Authors |
||
|
Comments 1: While Table 2 presents comparative metrics, the model “spaCI” is described as demonstrating “superior performance” without clearly specifying the baseline models or datasets used for comparison, weakening the evaluative rigor. Greater transparency on cohort characteristics and evaluation context is needed.
|
||
|
Response 1: Thank you for pointing this out. We have revised Table 2 to now include “baseline comparisons” column which states the baseline models used for comparison
|
||
|
Comments 2: In section 4.1, the description of MERFISH and Visium integration lacks specificity: “integration often requires feature selection + imputation” should elaborate on which imputation techniques are most commonly applied and what their limitations are in cross-modality spatial resolution alignment.
|
||
|
Response 2: We appreciate this important point. The “integration considerations” column is now revised to imputation techniques and their limitations
Comment 3: The phrase “differentiation predictions[98]” should be corrected to “differentiation prediction models [98]” for grammatical agreement and clarity.
Response 3: Thank you for pointing this out, this phrase is now corrected for grammatical clarity.
Comment 4: The sentence “The atlas has become an essential resource...” lacks a citation immediately following and should be anchored to source [28].
Response 4: We have added a citation to this sentence.
|
||
|
4. Response to Comments on the Quality of English Language We have polished the manuscript to improve style, flow, and sentence variety per your valuable suggestions. |
||
|
5. Additional clarifications |
||
|
We trust that these revisions address the reviewers’ concerns thoroughly. We remain available to provide any further clarification or additional information requested. |
||
